# Regular walking exercise prior to knee osteoarthritis reduces joint pain in an animal model

**Junya Sakamoto**[1,2]*, **Syouta Miyahara**[3], **Satoko Motokawa**[4], **Ayumi Takahashi**[1,2],
**Ryo Sasaki**[2,5], **Yuichiro Honda**[1,2], **Minoru Okita**[1,2]

**1** Institute of Biomedical Sciences (Health Sciences), Nagasaki University, Nagasaki, Japan, **2** Department of Physical Therapy Science, Nagasaki University Graduate School of Biomedical Sciences, Nagasaki, Japan, **3** Department of Rehabilitation Medicine, Japan Organization of Occupational Health and Safety, Spinal Injuries Center, Fukuoka, Japan, **4** Department of Clinical Services, Nagasaki Rehabilitation Hospital, Nagasaki, Japan, **5** Department of Rehabilitation, Juzenkai Hospital, Nagasaki, Japan

☯ These authors contributed equally to this work.
* jun-saka@nagasaki-u.ac.jp

## Abstract

We investigated the effect of regular walking exercise prior to knee osteoarthritis (OA) on pain and synovitis in a rat monoiodoacetic acid (MIA)-induced knee OA model. Seventy-one male Wistar rats were divided into three groups: (*i*) Sedentary + OA, (*ii*) Exercise + OA, and (*iii*) Sedentary + Sham groups. The Exercise + OA group underwent a regular treadmill walking exercise at 10 m/min (60 min/day, 5 days/week) for 6 weeks, followed by a 2-mg MIA injection in the right knee. The right knee joint was removed from rats in this group at the end of the 6-week exercise period and at 1 and 6 weeks after the MIA injection. After the 6 weeks of treadmill exercise but before MIA injection, there were no significant differences among the three groups in the pressure pain threshold, whereas at 1 week post-injection, the Exercise + OA group's pressure pain threshold was significantly higher than that in the Sedentary + OA group, and this difference persisted until the end of the experimental period. The histological changes in articular cartilage and subchondral bone revealed by toluidine blue staining showed no difference between the Sedentary + OA and EX + OA groups. The expression levels of interleukin (IL)-4 and IL-10 mRNA in the infrapatellar fat pad and synovium were significantly increased by the treadmill exercise. Significant reductions in the number of CD68-, CD11c-positive cells and IL-1β mRNA expression and an increase in the number of CD206-positive cells were observed at 1 week after the MIA injection in the Exercise + OA group compared to the Sedentary + OA group. These results suggest that regular walking exercise prior to the development of OA could alleviate joint pain through increases in the expressions of anti-inflammatory cytokines in the rat infrapatellar fat pad and synovium.

## Introduction

Exercise is a noninvasive and useful treatment for musculoskeletal pain [1], and many epidemiological surveys have established that middle-aged and elderly people who regularly engage

**Data Availability Statement:** All relevant data are within the paper and its Supporting information files.

**Funding:** This work was supported by a JSPS KAKEN Grant-in-Aid for Scientific Research (C) Number 19K11347 from the Ministry of Education, Science, Sports and Culture (MEXT), from 2019 to 2022 and Grant-in-Aid for Scientific Research (B) Number 22H03455 from the Ministry of Education, Science, Sports and Culture (MEXT), from 2022 to 2023. The funder had no role in study design, data collection and analysis, decision to publish, or preparation of the manuscript.

**Competing interests:** The authors have declared that no competing interests exist.

in an exercise habit have a significantly reduced risk of developing musculoskeletal pain [2–4]. Thus, exercise can be one of the preventive approaches for chronic musculoskeletal pain conditions. Basic studies of animal models have demonstrated the effect of exercise on reducing some pain conditions and their biological mechanisms [5–9]. For example, a regular wheel-running exercise before a chronic constriction injury prevented the full development of allodynia; the accumulation of M1 macrophages was suppressed and that of M2 macrophages was increased at the injury site by exercise [6]. Regular treadmill-running exercise in normal rats increased the expressions of interleukin (IL)-4 and IL-10 in synoviocytes of the knee joint [10]. IL-4 is known to have the effect of differentiating resident macrophages and M1 macrophages into M2 macrophages [11]. It has also been shown that IL-10 suppresses the accumulation of macrophages in inflamed tissues [12]. Thus, exercise may reduce inflammation even when peripheral tissues are damaged, which results in mild pain.

Knee osteoarthritis (OA) is a common chronic joint disease worldwide, with an incidence that is predicted to rise due to the aging population [13]. The biological mechanism of knee OA pain has been clarified. Synovitis caused by the phagocytosis of cartilage debris created by the wear and degeneration of articular cartilage is related to joint pain [14, 15]. It has been reported that the severity of synovitis on magnetic resonance imaging (MRI) correlates with the severity of pain [16]. Synovitis is thus one of the targets for pain control in knee OA. If synovitis can be suppressed even when the joint cartilage is worn, the pain caused by knee OA may be mild, and based on the aforementioned findings that regular exercise prior to developing of some pain conditions alleviate pain by lessening development of inflammation and can be one of the preventive approaches for chronic musculoskeletal pain conditions. Thus, we hypothesized that regular exercise prior to knee OA will make the pain induced milder. We investigated (*i*) the effect of regular walking exercise prior to knee OA on pain after the onset of knee OA and (*ii*) its mechanism, using a rat monoiodoacetic acid (MIA)-induced knee OA model.

## Materials and methods

All protocols for animal procedures were reviewed and approved by the Ethics Review Committee for Animal Experimentation at Nagasaki University (No. 1808091472–9). All experimental procedures were performed with the rats under anesthesia to ease or prevent suffering. A diagram of the experiment is shown in Fig 1. There are no financial or non-financial conflicts of interest related to this study.

### Animals

Seven-week-old male Wistar rats (CLEA Japan, Tokyo) were maintained at the Nagasaki University Center for Frontier Life Sciences. The rats were maintained in $30 \times 40 \times 20$-cm cages (two rats/cage) and exposed to a 12-h light-dark cycle at an ambient temperature of 25˚C. Food and water were provided ad libitum. Seventy-one rats were included in the study.

We divided the 71 rats into the following three groups. (1) The Sed + OA group (n = 20) in which knee OA was induced after normal storing and feeding (i.e., sedentary condition [Sed]) for 6 weeks, followed by normal breeding. (2) The Ex + OA group (n = 26) in which regular treadmill walking exercise (Ex) was applied for 6 weeks (wks) before knee OA was induced, followed by normal storing and feeding. (3) The Sed + Sham group (n = 25) in which normal storing and feeding was applied for 6 wks before a saline injection was administered as a sham treatment, followed by normal breeding. Rats that showed weight loss during the experiment were excluded from the experiment. The rats were randomly divided into the three groups

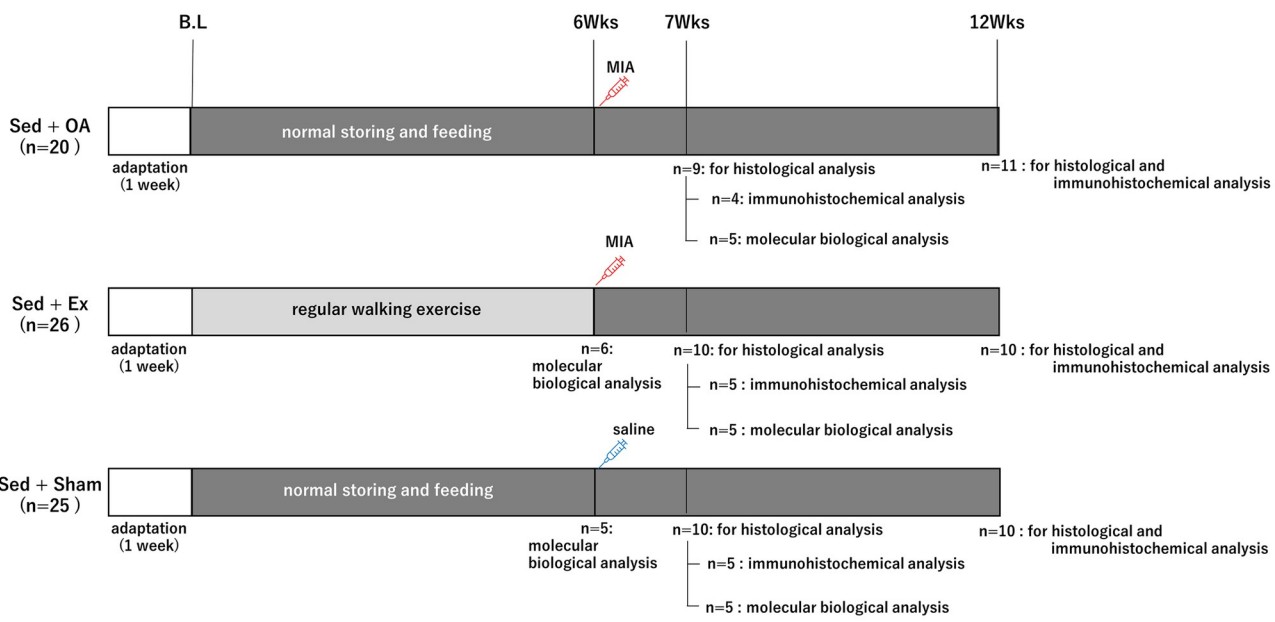

**Fig 1. The time course of the experiment.** Ex: exercise, OA: osteoarthritis, Sed: sedentary.

with the use of randomly generated serial numbers in Excel. Potential confounders such as the order of treatments and measurements were not controlled.

## Walking exercise with treadmill

Bedford et al. investigated the rate of exercise and oxygen uptake when rats were subjected to treadmill exercise, and they reported that the rates were 29.8% of the maximum oxygen uptake at 10 m/min, 46.7% at 15 m/min, and 63.8% at 20 m/min [17]. In humans, it is known that the light intensity of endurance-type activity is 20%–39% of the maximal oxygen uptake [18]. We used the treadmill exercise speed 10 m/min in the present study to simulate normal human walking.

The rats in the Ex + OA group were subjected to walking exercise at 10 m/min and 0˚ inclination on a small animal treadmill (Shinano Seisakusho, Tokyo) for 60 min/day, 5 days/wk for 6 wks. In addition, considering the effect on pain of the rats' stress due to entering the treadmill device, the OA group and the sham group were left in the treadmill device with the power turned off for 60 min/day. For their acclimatization to the environment inside the device, all rats were left in the treadmill device for 30 min/day with the power turned off beginning 1 wk before the start of the experiment.

In this study, from the perspective of primary prevention, to investigate the effect of regular walking exercise prior to knee OA on pain after the onset of OA, we did not subject the rats to the walking exercise after the injection of MIA.

## Intra-articular injection of mono-iodoacetate

Monosodium iodoacetate (MIA) causes joint pathology via the inhibition of glycolysis, thereby causing chondrocyte death, classified as secondary OA, and a 2-mg dose of MIA produced robust, reproducible joint pain with no effect on the general health of the animals [19]. In the present study, all of the rats that received the MIA or saline injection were anesthetized with

the combination of 0.375 mg/kg medetomidine (Kyoritu Pharma, Tokyo), 2.0 mg/kg midazolam (Sandoz Pharma Co., Tokyo), and 2.5 mg/kg butorphanol (Meiji Seika Pharma, Tokyo). With the use of a 31-gauge needle, 25 μL of sterile saline with 2 mg of MIA (Sigma-Aldrich, St. Louis, MO, USA) was injected into the right knee joint of each rat, via the patellar ligament. Rats in the Sed + Sham group (in which OA was not induced) were injected with 25 μL of saline in a similar way, as a sham treatment. This procedure was performed by JS alone and concealed from the other study authors. After the injection of MIA or saline into the joint, if the distress caused to the animal due to infection of the affected area or other unforeseen circumstances became significantly aggravated, the experiment was terminated in order to relieve the animal from the distress, and the animal was euthanized by an application of inhaled carbon dioxide.

## Pain behavioral assessment

During the experiment, we measured the pressure pain threshold (PPT) of the right knee joint. This measurement was taken at the start of the experiment, at 6 wks after the start of the experiment, at 1 wk after the MIA or saline injection (Sed + OA group, n = 20; Ex + OA group, n = 20; Sed + Sham group, n = 20), and 8–12 wks after the MIA or saline injection (Sed + OA group, n = 11; Ex + OA group, n = 10; Sed + Sham group, n = 10). A 1-wk adaptation period was provided prior to the start of the experiment.

The PPT was measured by two evaluators who were blinded to the groups to which the rats belonged. The PPT of the right knee joint was assessed using a Randall-Selitto apparatus (Ugo Basile, Varese, Italy). For the PPT assessment, the rat was placed in a sock with its hind paws protruding from the sack, and it was restrained by the evaluator's hand. The round tip of the transducer probe of the apparatus (base dia. 9 mm) was applied to the lateral side of the rat's right knee joint with increasing pressure (48 g/s). The threshold was defined as the force required to elicit either the hind-limb flection reflex or vocalization. Five measurements were taken at intervals of $\geq$3 min, and the average was recorded as the PPT, excluding the maximum and minimum measurements.

## Tissue sampling and preparation

At the end of the exercise period, before the injection of MIA or saline, rats in the Sed + Sham (n = 5) and Ex + OA (n = 6) groups were anesthetized, and the infrapatellar fat pad and medial and lateral synovium of the rats' right knee joints was treated with RNAlater® reagent (Ambion, Carlsbad, CA) immediately after excision for use in the molecular biological analysis.

Next, at 1 wk after the injection of MIA or saline, five rats from each group (Sed + OA, n = 5; Ex + OA, n = 5; Sed + Sham, n = 5) were anesthetized, the infrapatellar fat pad and medial and lateral synovium was harvested in the manner described above, and store at -80˚C until use in the molecular biological analysis. The right knee was subsequently removed from each group of rats (Sed + OA, n = 9; Ex + OA, n = 10; Sed + Sham, n = 10), including those from which the infrapatellar fat pad and synovium, were removed after transcardial perfusion with saline plus 4% paraformaldehyde dissolved in 0.01 M phosphate buffer, pH 7.4. The right knee joint was decalcified with 10% ethylenediaminetetraacetic acid in 0.01 M phosphate buffer, pH 7.4. Each specimen was embedded in paraffin.

At the end of the experiment, rats from each group (Sed + OA, n = 11; Ex + OA, n = 10; Sed + Sham, n = 10) were anesthetized, and the right knee joint of each rat was then removed as described above and samples were prepared as described above.

## Histological evaluation of the knee joints

From each paraffin-embedded knee joint sample, two 5-μm-thick frontal sections were made every 100 microns with a microtome and subjected to toluidine blue staining (at At 7 wks; Sed + OA, n = 9; Ex + OA, n = 10; Sed + Sham, n = 10, at 12wks; Sed + OA, n = 11; Ex + OA, n = 10; Sed + Sham, n = 10). The lesions of cartilage and subchondral bone were then examined microscopically under a light microscope (ECLIPSE 50i, Nikon, Tokyo) and scored using the Osteoarthritis Research Society International (OARSI) grading system criteria [20]. For the cartilage degeneration score, the tibial plateau on each section was divided into three zones of equal width, and the cartilage degeneration in each zone was scored from 0 (best) to 5 (worst). The total cartilage degeneration score was calculated by adding the values obtained for each zone. The subchondral bone damage score was determined on a numerical scale from 0 (best) to 5 (worst). The most severe lesion of the tibial plateau in each section was scored. The individual performing the analysis was blind to the groups to which the rats belonged.

## Analysis of macrophages in the synovium of the right knee joint

Two frontal 5-μm-thick sections from each rat (at At 7 wks; Sed + OA, n = 4; Ex + OA, n = 5; Sed + Sham, n = 5, at 12wks; Sed + OA, n = 11; Ex + OA, n = 10; Sed + Sham, n = 10) were subjected to an antigen retrieval step by incubation in 0.01 M citrate buffer (pH 6.0) followed by a 30-min incubation at room temperature with 0.6% $H_2O_2$ dissolved in methanol. The sections were blocked with 1% bovine serum albumin in phosphate-buffered saline containing 5% normal horse serum for 60 min, which provided better blocking compared to a single application of normal horse serum. The same sections were then incubated with a mouse anti-CD68 monoclonal antibody (1:500; Bio-Rad, Hercules, CA), rabbit anti-ITGAX/CD11c polyclonal antibody (1:250; LifeSpan Biosciences, Seattle, WA), or rabbit anti-mannose receptor polyclonal antibody (1:3000; Abcam, Cambridge, MA) overnight at room temperature, followed by incubation with biotinylated horse anti-mouse IgG (BA-2000; 1:2000; Vector Laboratories, Burlingame, CA) or biotinylated goat anti-rabbit IgG (H + L) (BA-1000; 1:2000; Vector Laboratories). Sections from at 12 wks sample were only subjected only for anti-CD68 monoclonal antibody.

Each section was stained using an avidin-biotin complex method (Vectastain Elite ABC kit; Vector Laboratories) and then visualized with a metal-enhanced DAB substrate kit (Thermo Fisher Scientific, Waltham, MA). Each section was stained with hematoxylin or methyl green. The medial and lateral synovium sections were photographed at 400× magnification with a digital camera. For all images, in the part of 100 μm from the synovial lining to the fibrous layer, the number of positive cells and the area were measured, and the number of positive cells/mm$^2$ was calculated. The individual performing the analysis was blind to the rats' groups.

## Molecular biological analysis

Total RNA was extracted from the infrapatellar fat pad and synovium samples (at At 7 wks; Sed + OA, n = 5; Ex + OA, n = 5; Sed + Sham, n = 5) with the use of an RNeasy Fibrous Tissue Mini Kit (Qiagen, Valencia, CA). However, enough RNA could not be extracted from one sample in the Sed + Sham group and was therefore excluded from the analysis. Total RNA was used as a template with a QuantiTect Reverse Transcription Kit (Qiagen) to prepare cDNA, and a real-time reverse transcription-polymerase chain reaction (RT-PCR) was performed using Brilliant III Ultra-Fast SYBR Green QPCR Master Mix (Agilent Technologies, Santa Clara, CA). The cDNA concentration of all samples was unified to 25 ng/μL, and 0.2 μL of the cDNA was applied to each well. The synthetic gene-specific primers are listed below in Table 1. The cycle threshold (Ct) was determined using an Mx3005P Real-Time QPCR System

**Table 1. Arrangement of synthetic gene-specific primers.**

| Object gene | F/R | Arrangement | GenBank no. |
|---|---|---|---|
| IL-4 | F | 5'-CCACGGAGAACGAGCTCATC-3' | AY496861.1 |
| | R | 5'-ACCGAGAACCCCAGACTTGTT-3' | |
| IL-10 | F | 5'-AAAGCAAGGCAGTGGAGCAG-3' | L02926.1 |
| | R | 5'-TCAAACTCATTCATGGCCTTGT-3' | |
| IL-1β | F | 5'-AATGACCTGTTCTTTGAGGCTGAC-3' | BC091141.1 |
| | R | 5'-CGAGATGCTGCTGTGAGATTTGAA-3' | |
| β-actin | F | 5'-GTGCTATGTTGCCCTAGACTTCG-3' | BC063166.1 |
| | R | 5'-GATGCCACAGGATTCCATACCC-3' | |

(Agilent Technologies). The mRNA expression of target genes was calculated using the ΔΔct method. Regarding IL-1β mRNA, the samples obtained 7 days after the MIA or saline injections were used.

## Statistical analyses

All data are presented as the mean ± standard deviation (SD). The rats' PPT values were assessed by a two-way analysis of variance (ANOVA), followed by the Bonferroni method. We applied the Kruskal-Wallis test for comparisons among the three groups for the cartilage degeneration score and subchondral bone degeneration score, and when significant differences were found, we applied the Steel-Dwass method as a post hoc test to determine significant differences among the three groups. The between-group differences in other parameters were assessed by a one-way ANOVA followed by the Bonferroni method. Differences were considered significant at $p < 0.05$.

## Results

### Between-group differences in the PPT

At the baseline, there were no significant differences in the PPT among the Sed + Sham, Sed + OA, and Ex + OA groups. Similarly, there were no significant differences in PPT among the three groups at 6 wks after the start of the experiment (before the injection of MIA or saline).

However, at 7 wks, i.e., 1 wk after the injection, and 12 wks after the start of the experiment, the PPT values in the Sed + OA and Ex + OA groups were significantly lower than those in the Sed + Sham group, and these differences persisted until the end of the experimental period. In comparison between the Sed + OA and Ex + OA groups, Ex + OA groups showed significantly higher values than the Sed + OA group. (Fig 2).

### Histological evaluation of the knee joints

At 7 wks after the start of the experiment, normal articular cartilage was observed in the Sed + Sham group, whereas in the Sed + OA and Ex + OA groups, the staining of cartilage matrix in the surface layer of articular cartilage was decreased (Fig 3A). The median cartilage degeneration score was 0 in the Sed + Sham group, 2 in the Sed + OA group, and 2 in the Ex + OA group, showing significantly higher values in the Sed + OA and Ex + OA groups versus the Sham group, with no significant difference between the two +OA groups (Fig 3B). In subchondral bone, only the usual findings were observed in all three groups.

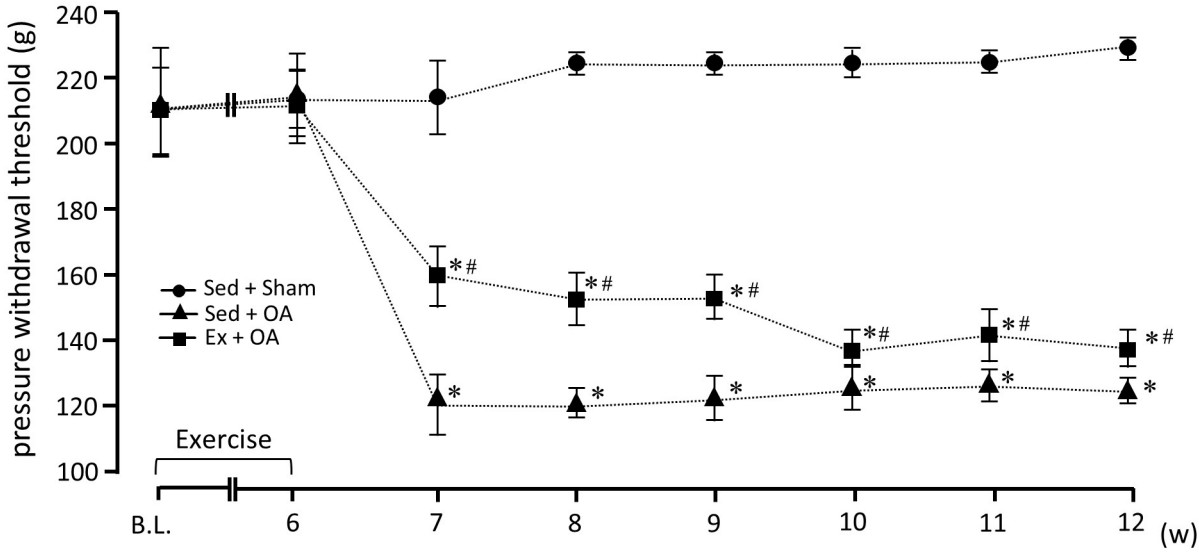

**Fig 2. Changes in the PPT of the right knee joint over 12 wks.** Data are mean ± SD. *p<0.05 vs. the Sed + Sham group. #p<0.05 vs. the Sed + OA group. Ex: exercise, OA: osteoarthritis, Sed: sedentary.

At the end of the experimental period, normal articular cartilage was observed in the Sed + Sham group, with no signs of degeneration. Normal histological findings were also observed in the Sed + Sham group's subchondral bone and bone marrow. In contrast, in the Sed + OA group and the Ex + OA group, cracks extending into the deep layer of the articular cartilage and a decrease in the staining properties of the cartilage matrix from the surface layer to the deep layer were observed. In addition, compensatory cartilage hyperplasia after bone tissue damage was observed in the subchondral bone where articular cartilage had disappeared, in both of these groups. Spindle-shaped fibroblast-like cells that are not normally found in the bone marrow cavity were also present (Fig 3A). The median cartilage degeneration score was 0 in the Sed + Sham group, 10 in the Sed + OA group, and 11 in the Ex + OA group. The scores of the Sed + OA and Ex + OA groups were significantly higher than that of the Sed + Sham group, with no significant difference between the Sed + OA and Ex + OA groups (Fig 3C).

The median subchondral bone degeneration score was 0 in the Sed + Sham group, 2 in the Sed + OA group, and 1 in the Ex + OA group. Here too, the scores of the Sed + OA and Ex + OA groups were significantly higher than that of the Sed + Sham group, with no significant difference between the two Ex groups (Fig 3D).

## Macrophages in the synovium of the right knee joint

At 7 wks after the experiment's start, the number of CD68-positive cells per unit area was 60.6 ± 13.0 cells/mm$^2$ (Sed + Sham), 139.0 ± 23.3 cells/mm$^2$ (Sed + OA), and 112.3 ± 13.6 cells/mm$^2$ (Ex + OA). The number of CD11c-positive cells per unit area was 40.8 ± 8.4 cells/mm$^2$ (Sed + Sham), 99.1 ± 10.3 cells/mm$^2$ (Sed + OA), and 63.1 ± 16.5 cells/mm$^2$ (Ex + OA). For both parameters, the Sed + OA and Ex + OA groups' values were significantly higher than those of the Sed + Sham group, and the Ex + OA group's values were significantly lower than those of the Sed + OA group (Fig 4B and 4C).

In contrast, the number of CD206-positive cells per unit area was 27.7 ± 5.9 cells/mm$^2$ in the Sed + Sham group, 34.9 ± 4.8 cells/mm$^2$ in the Sed + OA group, and 73.3± 15.4 cells/mm$^2$

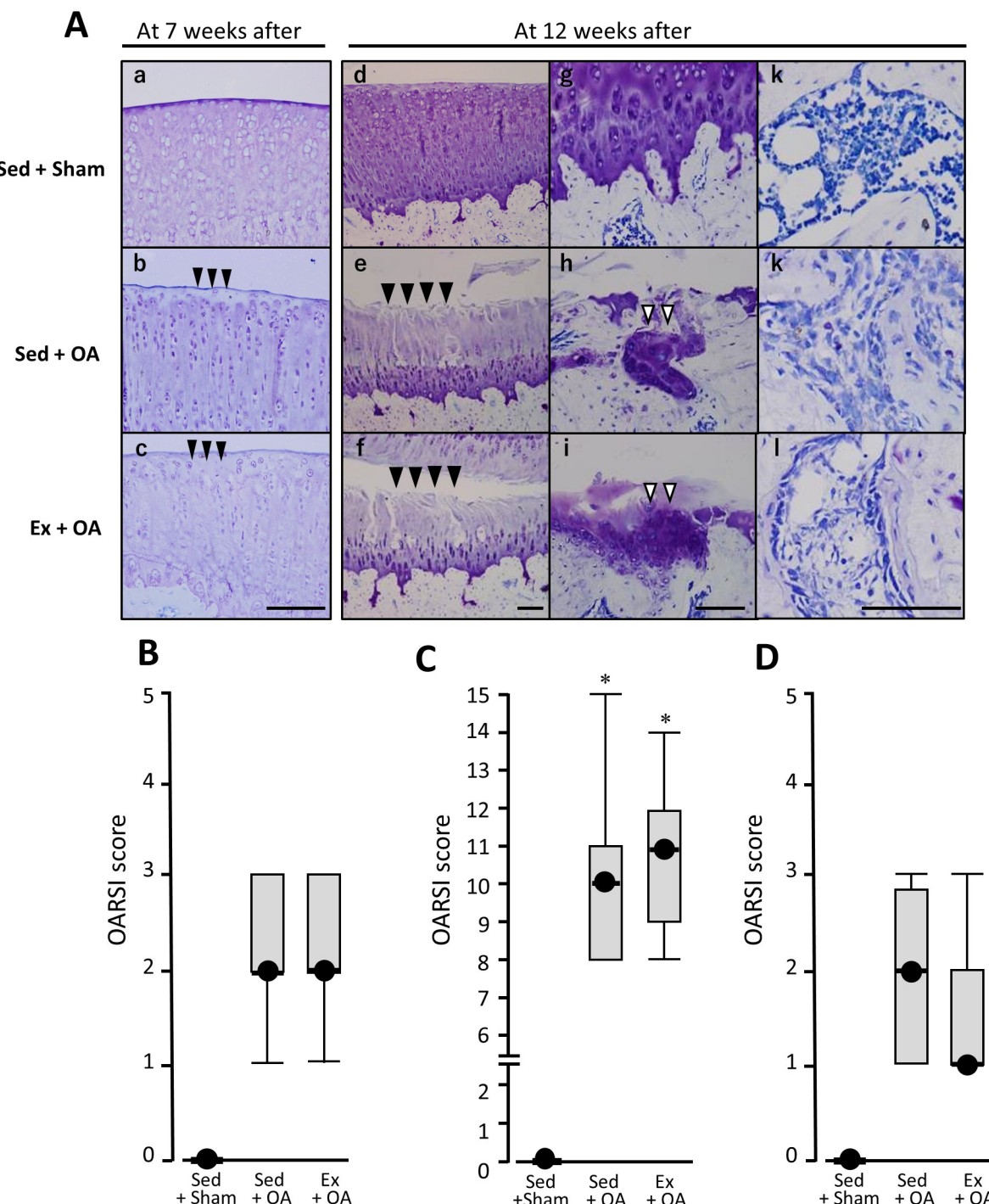

**Fig 3. Cartilage and subchondral bone degeneration scores at 7 and 12 wks after the start of the experiment. A:** a, b and c; Representative toluidine blue staining of each group at 7 wks after the start of experiment. In b and c, *Black arrowheads* indicate the decrease of the staining of cartilage matrix in the superficial layer of articular cartilage. Scale bar, 50 μm. d, e, f, g, h, I, j, k and l; At 12 wks after the start of the experiment: *black arrowheads* indicate cracks in articular cartilage, and *white arrowheads* indicate compensatory cartilage hyperplasia (second from left, middle and bottom). Scale bar, 100 μm. **B:** OARSI cartilage degeneration score at 7 wks after the start of the experiment. **C:** OARSI cartilage degeneration score at 12 wks. **D:** OARSI subchondral bone degeneration score at 12 wks. Data are mean ± SD. *p<0.05 vs. the Sed + Sham group. #p<0.05 vs. the Sed + OA group. Ex: exercise, OA: osteoarthritis, Sed: sedentary.

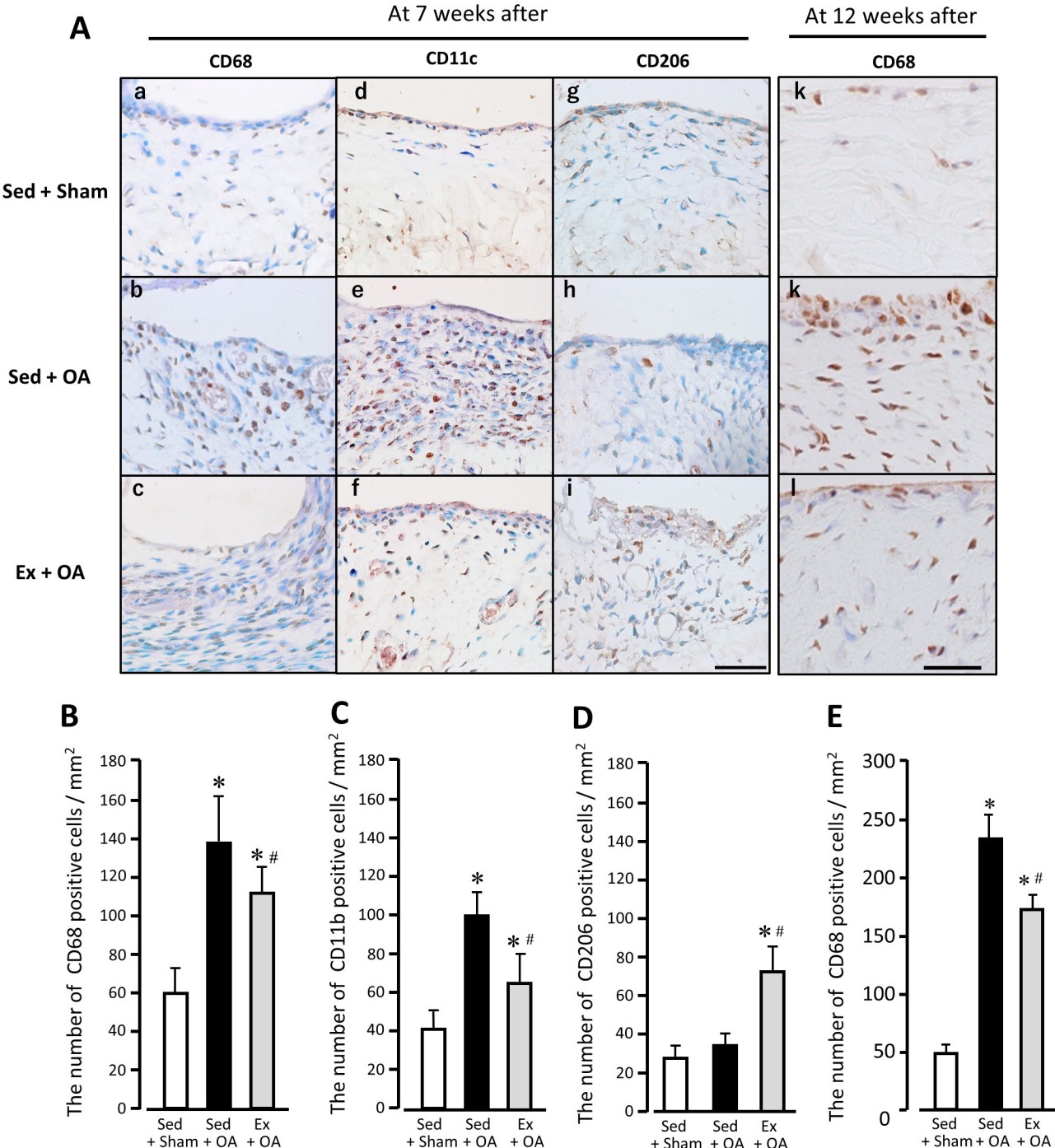

**Fig 4. Immunohistochemical findings (A) and the number of CD68- (B, E), CD11c- (C), and CD206- (D) positive cells in synovium of the right knee joint at 7 and 12 wks after the start of the experiment.** (A) Representative immunohistochemical staining for CD68 (*left*), CD11c (*2nd from left*) and CD206 (*2nd from right*) in synovium at 7 wks after the start of the experiment, Scale, 50 μm. CD68 (*right*) at 12 wks. Scale bar, 25 μm. Data are mean ± SD. *p<bar 0.05 vs. the Sed + Sham group. #p<0.05 vs. the Sed + OA group. The number of CD68- (B), CD11b- (C), and CD206- (D) positive cells in synovium in each group at 7 wks. The number of CD68 I positive cells in synovium in each group at 12 wks. Data are mean ± SD. *p<0.05 vs. the Sed + Sham group. #p<0.05 vs. the Sed + OA group. Ex: exercise, OA: osteoarthritis, Sed: sedentary.

in the Ex + OA group, the latter of which was significantly higher than those of the other two groups (Fig 4D).

At the end of the experimental period, the number of CD68-positive cells per unit area was $48.1 \pm 8.0$ cells/mm$^2$ in the Sed + Sham group, $230.7 \pm 31.6$ cells/mm$^2$ in the Sed + OA group, and $173.7 \pm 11.5$ cells/mm$^2$ in the Ex + OA group, with significantly higher values in the Sed + OA and Ex + OA groups compared to the Sed + Sham group, and a significantly lower value in the Ex + OA group compared to the Sed + OA group (Fig 4E).

## Changes of IL-4, IL-10, and IL-1β mRNA in the infrapatellar fat pad and synovium

To investigate the effects of the treadmill walking exercise on the expression of anti-inflammatory cytokines in the rats' infrapatellar fat pad and synovium, we obtained samples at the end of the 6 wks of treadmill walking (before the injection) from the Sed + Sham and Ex + OA groups. After the 6 wks' exercise, The IL-10 mRNA expression in the infrapatellar fat pad and synovium was $0.83 \pm 0.30$ in the Sed + Sham group and $1.81. \pm 0.29$ in the Ex + OA group, showing a significantly higher level in the exercise group than in the non-exercise group (Fig 5B).

Next, to investigate the IL-4 and IL-10 mRNA expressions after the injection of MIA or saline, samples were obtained at 1 wk post-injection. The IL-4 mRNA expression levels were $1.23 \pm 0.19$ (Sed + Sham), $0.24 \pm 0.08$ (Sed + OA), and $1.19 \pm 0.27$ (Ex + OA), with the Sed + OA group showing significantly lower levels than the Sed + Sham group. On the other hand, the Ex + OA group showed significantly higher values than the Sed + OA group, and there was no significant difference between the Ex + OA and Sed + Sham groups (Fig 5C). The IL-10 mRNA expression was $0.85 \pm 0.23$ (Sed + Sham), $0.29 \pm 0.19$ (Sed + OA), and $0.68 \pm 0.16$ (Ex + OA), showing significantly lower values in the Sed + OA groups compared to the Sed + Sham group. In contrast, the Ex + OA group showed significantly higher values than the Sed + OA group, and there was no significant difference between the Ex + OA and Sed + Sham groups (Fig 5D).

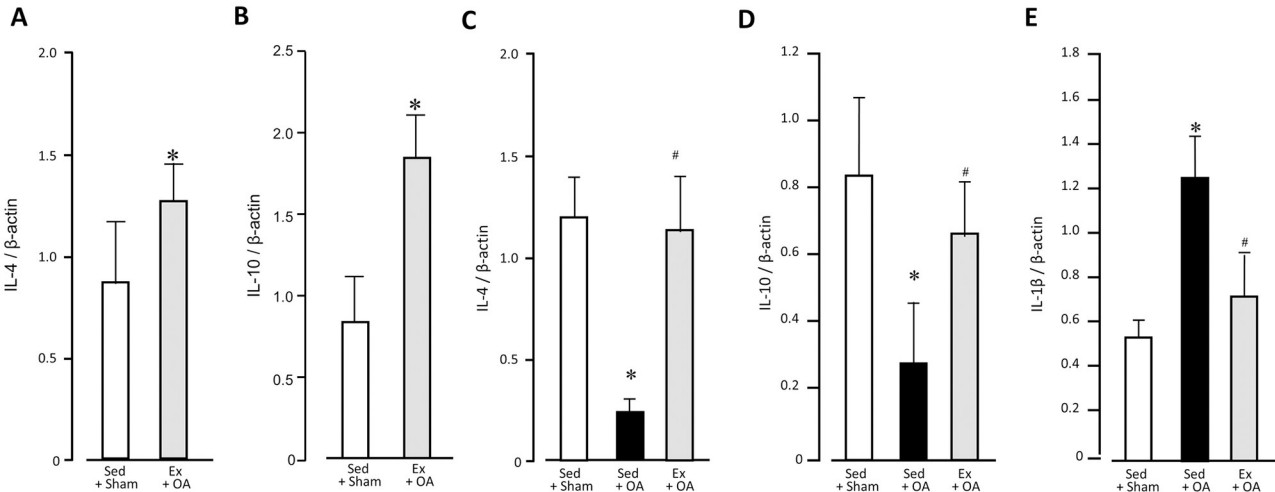

**Fig 5. Changes of IL-4 mRNA, IL-10 mRNA, and IL-1β mRNA after 6 wks of daily treadmill walking and 7 wks after the start of the experiment.** The expressions of IL-4 mRNA (**A**) and IL-10 mRNA (**B**) in the infrapatellar fat pad and synovium at 6 wks after the start of the experiment, and the expressions of IL-4 mRNA (**C**) and IL-10 mRNA (**D**) at 7 wks. **E:** The expression of IL-1β mRNA at 7 wks. Data are mean ± SD. *p<0.05 vs. the non-Ex group. #p<0.05 vs. the Sed + OA group. Ex: exercise, OA: osteoarthritis, Sed: sedentary.

The expression levels of IL -1β mRNA were 0.51 ± 0.10 (Sed + Sham), 1.24 ± 0.23 (Sed + OA), and 0.73 ± 0.19 (Ex + OA), showing significantly higher levels in the Sed + OA group versus the Sed + Sham group. The Ex + OA group's values were significantly lower than those of the Sed + OA group, and no significant difference was detected between the Ex + OA and Sed + Sham groups (Fig 5E).

## Discussion

We performed histological, immunohistochemical, biochemical, and molecular biological analyses to investigate the effects of regular walking exercise conducted before the onset of knee OA in rats on the pain that was induced by OA, and we evaluated the underlying biological mechanisms. The PPT results in our study showed that regular walking exercise before the onset of knee OA may palliate pain. Basic research using rat models has also shown that pain induced after regular exercise is less severe than pain induced without exercise [4–8]. Our present PPT findings are similar to those of these previous studies, suggesting that regular exercise may be useful for the primary prevention of chronic pain induced by knee OA.

The degeneration of articular cartilage and subchondral bone is a characteristic histological change in knee OA, which is one aspect of the pathology of knee OA pain [21]. We observed similar histological findings in the Sed + OA and Ex + OA groups at both 1 and 6 wks after the OA challenge, with no significant between-group differences in the cartilage or subchondral degeneration scores. The histological changes that are characteristic of knee OA were progressive after the MIA injection in both OA groups. The rat OA model in this study is thus valid, and the prior regular walking exercise did not alleviate the degeneration of the articular cartilage or subchondral bone after the onset of knee OA, suggesting that other factors contribute to the mechanism by which the pain after the onset of OA is reduced. However, the rats in the Ex + OA group did not continue to exercise, and it is thus not clear how the degeneration of articular cartilage and subchondral bone progresses when regular walking exercise is continued after the onset of OA, which is one of the limitations of this study.

We observed a decrease in the number of total and M1 macrophages and an increase in the number of M2 macrophages in the Ex + OA group compared to Sed + OA group. In an OA synovitis, it is known that macrophages accumulate in the synovium and that inflammatory M1 macrophages increase and anti-inflammatory M2 macrophages decrease [22]. An earlier study has been reported that an increase in the number of macrophages in the synovium leads to an upregulation of inflammatory mediators, causing severe synovitis [23]. Our results thus suggest that regular walking exercise may suppress the accumulation of macrophages in the synovium after the onset of knee OA and promote the polarization from M1 to M2 macrophages; it is possible that a suppression of synovitis is involved in the mechanism by which the PPT after the onset of knee OA was mild in the present Ex + OA group.

The synovium and infrapatellar fat pad are parts of the same joint components, and the alteration of cytokines in the infrapatellar fat pad plays a deleterious role in OA because of its close interaction with synovium in the same functional unit [24, 25]. Our molecular biological analysis findings suggest that regular walking exercise may increase the expressions of IL-4 and IL-10 in the knee joint. IL-4 is known to induce differentiation from resident macrophages and M1 macrophages to M2 macrophages [11]. These phenotypic changes in macrophages are important responses toward the convergence of inflammatory responses [26]. In addition, IL-10 has been shown to suppress the expression of IL-1β produced by M1 macrophages [12, 27]. Based these previous findings, our results may indicate that regular walking promotes a favorable biological environment that can reduce inflammation even when it occurs. In fact, in the Ex + OA group, the expression level of IL-1β mRNA at 1 wk after the MIA injection was

significantly lower than that in the Sed + OA group. It is known that IL-1β induces pain by stimulating primary nociceptive neurons in OA synovium [28]. Therefore, in the present Ex + OA group, the accumulation of macrophages in the synovium after the onset of knee OA was suppressed by the actions of IL-4 and IL-10, and the differentiation from M1 to M2 macrophages was promoted, leading to a decrease in the expression of IL-1β, suggesting that the decrease in the PPT in the right knee joints after the induction of knee OA was mild.

Chronic pain caused by knee OA not only leads to prolonged impairment and dysfunction; it also leads to a decline in an individual's quality of life (QOL) [29] and socioeconomic loss [30]. Our present results provide important basic data showing that the practice of regular walking is useful as a primary prevention strategy for chronic knee OA pain. The walking exercise investigated in this study is a widely practicable exercise even for middle-aged and elderly people, and it can be included in health guidance for populations who are at high risk of developing knee OA, which is one of the causative diseases of chronic knee pain. A continued accumulation of basic data is necessary to further verify the effects of regular exercise by humans as a primary prevention strategy for pain, reductions in QOL, and medical economic costs.

## Study limitations

In this study, the treadmill exercise speed is determined according to previous studies [17, 18], and cardiopulmonary exercise testing was not measured in this study. In the next studies, we need to perform a graded load test until exhaustion and determine the maximum aerobic power in individual rats based on the maximum speed reached in the test. Next, the dynamics of cytokines in the rat infrapatellar fat pad and synovium have not been analyzed at the protein level. Third, we were not able to examine the numbers of macrophages after the treadmill walking period. it is not clear whether M2 macrophages increase occurred before or after the onset of knee OA. We were also not able to examine chemotactic factors such as monocyte chemoattractant protein (MCP)-1 that promote macrophage accumulation. In a study using a mouse model of neuropathic pain, when neuropathic pain was induced after treadmill running, a decrease in MCP-1 expression and a decrease in macrophages were observed at the injury site [5]. The mechanism by which the accumulation of macrophages after the onset of knee OA was suppressed in the present investigation may be related to MCP-1, which is also a subject for further investigation. In addition, pain-related molecules other than IL-1β were not investigated in this study. In knee OA, pain is triggered by inflammatory mediators such as IL-6, tumor necrosis factor-alpha (TNF-α), and nerve growth factor in the synovium [31]. By examining these mediators, we may be able to obtain higher-quality evidence regarding the effects of regular walking exercise on knee osteoarthritis.

## Conclusion

We assessed the effect of regular treadmill walking on pain after the onset of knee OA and its biological mechanism. The results revealed that 6 wks of treadmill walking before the onset of knee OA may palliate pain caused by knee OA. The expression levels of IL-4 and IL-10 mRNA may be associated with the decrease in the accumulation and polarization of macrophages in the rat infrapatellar fat pad and synovium, followed by a suppression of the expression level of IL-1β mRNA. These alterations may be involved in minor pain in the MIA injection-induced rat knee OA model. We were able to show the effects of regular walking even in degenerative joint diseases such as knee OA, in which an elucidation of the disease progression from the onset may be important for further clarification of the effects of preventive interventions for chronic pain.

## Supporting information

**S1 Data.**
(XLSX)

## Acknowledgments

We would like to thank KN international INC (https://www.kninter.co.jp/) for English language editing.

## Author Contributions

**Conceptualization:** Yuichiro Honda, Minoru Okita.

**Data curation:** Junya Sakamoto, Yuichiro Honda, Minoru Okita.

**Formal analysis:** Junya Sakamoto.

**Funding acquisition:** Junya Sakamoto.

**Investigation:** Junya Sakamoto, Syouta Miyahara, Satoko Motokawa, Ayumi Takahashi, Ryo Sasaki.

**Methodology:** Junya Sakamoto, Syouta Miyahara, Satoko Motokawa, Ayumi Takahashi, Ryo Sasaki.

**Project administration:** Junya Sakamoto, Minoru Okita.

**Resources:** Minoru Okita.

**Supervision:** Junya Sakamoto, Yuichiro Honda, Minoru Okita.

**Validation:** Junya Sakamoto, Syouta Miyahara, Satoko Motokawa, Ayumi Takahashi, Ryo Sasaki, Yuichiro Honda, Minoru Okita.

**Writing – original draft:** Junya Sakamoto.

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
