## [Decision Letter · Decision Letter 0]

3 Apr 2023

PONE-D-23-07563Regular walking exercise prior to knee osteoarthritis reduces joint pain and synovitis through an upregulation of anti-inflammatory cytokines in the jointPLOS ONE

Dear Dr. Sakamoto,

Thank you for submitting your manuscript to PLOS ONE. After careful consideration, we feel that it has merit but does not fully meet PLOS ONE’s publication criteria as it currently stands. Therefore, we invite you to submit a revised version of the manuscript that addresses the points raised during the review process.

We look forward to receiving your revised manuscript.

Kind regards,

Tanja Grubić Kezele, Ph.D., M.D.

Academic Editor

PLOS ONE

Journal Requirements:

2. To comply with PLOS ONE submissions requirements, in your Methods section, please provide additional information regarding the experiments involving animals and ensure you have included details on (1) methods of sacrifice, (2) efforts to alleviate suffering and (3) as part of your revision, please complete and submit a copy of the Full ARRIVE 2.0 Guidelines checklist, a document that aims to improve experimental reporting and reproducibility of animal studies for purposes of post-publication data analysis and reproducibility: https://arriveguidelines.org/sites/arrive/files/documents/Author%20Checklist%20-%20Full.pdf Please include your completed checklist as a Supporting Information file. Note that if your paper is accepted for publication, this checklist will be published as part of your article.

   "Initials of the authors who received each award: JS

Grant numbers awarded to each author: Grant-in-Aid for Scientific Research (C) Number 19K11347, from 2019 to 2022.

The full name of each funder: Japan Society for the Promotion of Science(JSPS) KAKEN from the Ministry of Education, Science, Sports and Culture (MEXT),

URL of each funder website: https://www.jsps.go.jp/j-grantsinaid/

Did the sponsors or funders play any role in the study design, data collection and analysis, decision to publish, or preparation of the manuscript?: No" 

Additional Editor Comments:

Based on the reviewers' suggestions, the paper needs major revision.  The reviewers' comments can be found below.

Reviewers' comments:

Reviewer's Responses to Questions

**Comments to the Author**

1. Is the manuscript technically sound, and do the data support the conclusions?

Reviewer #1: Yes

Reviewer #2: Partly

2. Has the statistical analysis been performed appropriately and rigorously? 

Reviewer #1: Yes

Reviewer #2: N/A

3. Have the authors made all data underlying the findings in their manuscript fully available?

Reviewer #1: Yes

Reviewer #2: No

4. Is the manuscript presented in an intelligible fashion and written in standard English?

Reviewer #1: Yes

Reviewer #2: No

5. Review Comments to the Author

Reviewer #1: Dear Authors,

in my opinion, the topic is interesting considering the large impact on health-related quality of life and cost of osteoarthritis in humans and accordingly the increased need to address to different strategies to address OA-related pain.

However, I have concerns about the methodological implant of the study and some critical issues should be addressed to improve the paper.

Major revisions

TITLE. I suggest specifying that the study is conducted on an animal model in the title, for example: “Regular walking exercise prior to knee osteoarthritis reduces joint pain and synovitis through an upregulation of anti-inflammatory cytokines in the joint of an animal model”.

MATERIALS AND METHODS. I suggest providing a time-line diagram, to increase understanding of the methodology.

MATERIALS AND METHODS. Please, clarify how animals were allocated to experimental groups.

MATERIALS AND METHODS. Please, characterize the operators who performed the data collection/analysis.

DISCUSSION. Future research scenarios should be highlighted, also considering possible human implications of this study. Osteoarthritis has disabling sequelae on the health-related quality of life and health-related cost, hence the therapeutic management has great implications on human life. According to this, you should cite the following references:

• de Sire A et al. Effectiveness of Combined Treatment Using Physical Exercise and Ultrasound-Guided Radiofrequency Ablation of Genicular Nerves in Patients with Knee Osteoarthritis. Applied Sciences. 2021; 11(10):4338. https://doi.org/10.3390/app11104338

• Törmälehto S, et al. Eight-year trajectories of changes in health-related quality of life in knee osteoarthritis: Data from the Osteoarthritis Initiative (OAI). PLoS One. 2019 Jul 19;14(7):e0219902. doi: 10.1371/journal.pone.0219902. PMID: 31323049; PMCID: PMC6641160.

• Espinoza MA, et al. Cost analysis of chronic pain due to musculoskeletal disorders in Chile. PLoS One. 2022 Oct 27;17(10):e0273667. doi: 10.1371/journal.pone.0273667. PMID: 36301984; PMCID: PMC9612497.

Minor revisions

INTRODUCTION. Page 4, line 75. Please, correct “more mild” with “milder”.

FIGURES and TABLES. An abbreviation list should be provided under each table and figures.

Reviewer #2: The idea of the manuscript is interesting; however, it needs revision on the English language by a native speaker. Also, it needs corrections in methods, results and discussion section. Further commets are in the attachment.

6. PLOS authors have the option to publish the peer review history of their article (what does this mean?). If published, this will include your full peer review and any attached files.

Reviewer #1: No

Reviewer #2: No

---

## [Author Response · Author response to Decision Letter 0]

25 May 2023

Tanja Grubić Kezele, Ph.D., M.D.

Academic Editor

We greatly appreciate your general and specific comments that clarified the problems of the manuscript, and we are thankful for your thoughtfulness and kindness.

We revised our manuscript as you pointed out. We deleted section of physical activity in the revised manuscript. We also made major revisions in line with the reviewer's advice. It also included corrections from native English proofreaders, which making it difficult to identify parts of corrections by Microsoft tracking function. For this reason, the line numbers in the revised manuscript of the revised parts in accordance with editor and reviewer advice were described. 

In submitting the revised manuscript, the following changes are made: in the process of creating the data set, there were errors in the numbers of some of the results mentioned in the first manuscript, thus, we corrected them. However, the results of statistical analysis did not change at all. One of coauthor (IT) was also excluded from the authors because one of them offered to decline.

Each of your comments have been answered below.

　We checked and corrected the format of our manuscript that you requested.

2. To comply with PLOS ONE submissions requirements, in your Methods section, please provide additional information regarding the experiments involving animals and ensure you have included details on (1) methods of sacrifice, (2) efforts to alleviate suffering and (3) as part of your revision, please complete and submit a copy of the Full ARRIVE 2.0 Guidelines checklist, a document that aims to improve experimental reporting and reproducibility of animal studies for purposes of post-publication data analysis and reproducibility:

 In line with your points, we described (1) methods of sacrifice () and humane endpoint (line 128-131) and (2) how to reduce suffering (line 78-79) in the Methods section. In addition, we also prepared Full ARRIVE 2.0 checklist.

3. Thank you for stating the following financial disclosure: "Initials of the authors who received each award: JS Grant numbers awarded to each author: Grant-in-Aid for Scientific Research (C) Number 19K11347, from 2019 to 2022. The full name of each funder: Japan Society for the Promotion of Science (JSPS) KAKEN from the Ministry of Education, Science, Sports and Culture (MEXT), URL of each funder website: https://www.jsps.go.jp/j-grantsinaid/ Did the sponsors or funders play any role in the study design, data collection and analysis, decision to publish, or preparation of the manuscript?: No" Please state what role the funders took in the study. If the funders had no role, please state: "The funders had no role in study design, data collection and analysis, decision to publish, or preparation of the manuscript."

 The funder, JSPS KAKEN from the Ministry of Education, Science, Sports and Culture (MEXT), had no role in study design, data collection and analysis, decision to publish, or preparation of the manuscript.

4. In your Data Availability statement, you have not specified where the minimal data set underlying the results described in your manuscript can be found. PLOS defines a study's minimal data set as the underlying data used to reach the conclusions drawn in the manuscript and any additional data required to replicate the reported study findings in their entirety. All PLOS journals require that the minimal data set be made fully available.

　We prepared and submitted the minimal data in accordance with your regulations.

Reviewer #1

We wish to express our strong appreciation to the reviewers for your insightful comments on our paper. We feel the comments have helped us significantly improve the paper. We revised our manuscript as you pointed out. Each of your comments have been answered below.

1. TITLE. I suggest specifying that the study is conducted on an animal model in the title, for example: “Regular walking exercise prior to knee osteoarthritis reduces joint pain and synovitis through an upregulation of anti-inflammatory cytokines in the joint of an animal model”.

You and reviewer #2 suggest that we correct the t it would be desirable to revise the title, we have revised it as follows: Regular walking exercise prior to knee osteoarthritis reduces joint pain in an animal model.

2. MATERIALS AND METHODS. I suggest providing a time-line diagram, to increase understanding of the methodology.

Your points are appropriate advice for the reader, and we made a time-line diagram of our experiment and made it Figure 1.

3. MATERIALS AND METHODS. Please, clarify how animals were allocated to experimental groups.

We appreciate your advice and apologize for not being able to adequately describe allocation. We added the description about allocation in line 88-92 and the number of rats in each group used in each analysis was also described.

4. MATERIALS AND METHODS. Please, characterize the operators who performed the data collection/analysis.

We added the characteristic of operators who collect and analyze data at the end of each analysis method section.

5. DISCUSSION. Future research scenarios should be highlighted, also considering possible human implications of this study. Osteoarthritis has disabling sequelae on the health-related quality of life and health-related cost, hence the therapeutic management has great implications on human life. According to this, you should cite the following references:

We appreciated your comment that make the DISCUSSION of our manuscript worthwhile. We added a description about what you have pointed out in the last paragraph of the discussion.

6. INTRODUCTION. Page 4, line 75. Please, correct “more mild” with “milder”.

I appreciate your point. We revised the manuscript in line 70 according to your advice.

7. FIGURES and TABLES. An abbreviation list should be provided under each table and figures.

I added a note about abbreviations in figure legends.

Reviewer #2:

We greatly appreciate your general and specific comments that clarified the problems of the manuscript, and we are thankful for your thoughtfulness and kindness.

We revised our first manuscript as you pointed out. Each of your comments have been answered below.

Introduction:

1. The idea of the manuscript is interesting; however, it needs revision on the English language by a native speaker.

When we submitted our first manuscript, we submitted it to a proofreading company and had it natively checked. However, it was a shame it wasn't up to the level you wanted. In submitting the revised manuscript, we conveyed your points and had it carefully checked again and issued a document certifying that the check had been made.

2. Title: “Regular walking exercise prior to knee osteoarthritis reduces joint pain and synovitis through an upregulation of anti-inflammatory cytokines in the joint”. In my opinion you cannot make the statement that the joint pain reduction was due to an upregulation of anti-inflammatory cytokines once you did not measured protein content, only mRNA expression of 3 different genes. The title should be something like “Regular walking exercise prior to knee osteoarthritis induction reduces joint pain”.

The title of the he first version of our manuscript was also pointed out by reviewer 1. Thus, based on your and reviewer 1's opinions, we modified the following: Regular walking exercise prior to knee osteoarthritis reduces joint pain in an animal model

3. I think the manuscript has few references, please include more references in the manuscript. 

Lines 48-49: reference

Lines 69-70: reference

About lines 48-49 (line 47 in revised manuscript) reference, we added the reference No.1, and we corrected the uncertain statements.

About lines 69-70 (line 64-65 in revised manuscript) reference, we added the reference No.14 and 15.

4. Lines 34-36: it needs to be clearer that there was no difference in the pressure pain threshold BEFORE the injections and that it changes AFTER the injections. It was a little bit confuse for me.

Following your advice, in lines 33 to 36 of the revised manuscript, wee corrected the text about the results of the pressure pain threshold.

5. Lines 44-46: walking exercise prior to OA alleviate… alleviate what? Pain? Clarify this on the abstract.

Thank you for pointing that out. We noted that regular walking exercise prior to the development of OA could alleviate “joint pain”.

6. Line 48: Use only exercise for all the sentences where appears physical activity and/or exercise.

We can avoid reader confusion by revising the manuscript according to your suggestions. We deleted “physical activity” and used only word “exercise”.

7. Line 77: “using a rat monoiodoacetic acid (MIA)-induced”. Why did you use this model? Please reference this model in this sentence.

As you noted, it is important to clarify this point. We added the reason for adopting a rat monoiodoacetic acid (MIA)-induced in line 118-120 in revised manuscript.

8. Why did you choose to start the exercise program before the monoiodoacetic acid injections and not after?

We believe that primary prevention is as important as secondary prevention for chronic musculoskeletal pain. Especially in countries with aging of society. The results of epidemiological studies suggest the effectiveness of exercise as a primary preventive management. However, basic research using animal models is essential to examine the mechanism as well, and in order to examine the primary preventive effect of exercise, it is necessary to conduct exercise only before the development of knee OA and to verify the effect. Therefore, exercise was loaded only before MIA injection in this study. But if, as you point out, you continue to exercise after a MIA, you may get better results compared to what you get now. This needs to be examined in the future. We added the reason for adopting exercise only before MIA injection in line 68-72 and line 115-117 in revised manuscript.

9. Why did you stop the exercise program after injections?

The answer to this question is the same as the answer to the previous question.

10. In my opinion, there is no two experiments. It is only the continuation of the same experiment. Therefore, do not divide them in two neither in the Methods nor in the Results.

We revised the manuscript in accordance with your opinion. The experiments were described together as one

Methods: 

11. Please specify the animal number for each analysis.

We appreciate your advice and apologize for not being able to adequately describe allocation. We added the description about allocation in line 88-92 and the number of rats in each group used in each analysis was also described, and we made a time-line diagram of our experiment and made it Figure 1.

12. Lines 99-100 “for the examination of the effect of continuous walking exercise on pain after the onset of knee OA”. It seems you will do the exercise after que onset of knee OA, however, you perform the exercise prior to the injections. It makes the experiment confuse, please reformulate to clarify this matter.

In line with your advice, we revised the method section, thus sentence you pointed out was deleted in the revised manuscript.

13. Lines 112-113: “We next examined the changes that occur within the knee joint due to loading regular walking exercises.” You actually examined the changes in the knee joint after OA induction with prior exercise or not. Please reformulate this sentence.

In line with your advice, we revised the method section, thus sentence you pointed out was deleted in the revised manuscript.

14. You always should, throughout the manuscript, clarify that the exercise was prior to the OA induction. For example in line 115: “regular walking exercise for 6 weeks” you should write “prior regular walking exercise for 6 weeks”. Because it can be confusion to the reader.

The text you pointed out was removed due to a major revision of the manuscript in the Methods section, but the finding you pointed out was utilized in the line 90 to 96.

15. Walking exercise with treadmill:

An observation, like humans, animals have a variable maximum oxygen uptake. Therefore, to make future studies more refined, you should perform an Incremental Load Test to every single rat and then you could say all of them walk at the same relative intensity. Because 10m/min could be 40% of the VO2max for one but 30% of the VO2max for another. Once you have the VO2max for every single rat, you separate the ones with similar VO2max and put them to run together.

As you point out, it is most desirable to determine the treadmill speed after measuring the maximum oxygen uptake of each rat. To modify that point, though, we need dedicated equipment, and we can't do it now. This point will be considered in the future.

16. Line 141: “Rats in the group that in which OA was not induced”. Please write the group name in all this case (sham group).

Sorry for the careless mistake. Correctly changed the sentence in line 128.

17. Assessment of physical activity:

Lines 156-158: “the measurements were obtained between 15:00 and 18:00, when the light-dark cycle in the rearing room was close to that of the light period and the rats' active period.” Please reference the statement that this is the rats’ active period. I think this result is irrelevant for this manuscript, once there is no difference between groups. Also, you cannot say based only in 1h tracking if the rat was more active or not due to the exercise. Therefore, this result should not be presented in the manuscript.

In light of your comments, we deleted the analysis and results related to physical activity.

18. Statistics: Why did you used One-way Anova to the other between-group parameters and not two-way anova?

Your question is reasonable because some analyses have two time points from which the data was taken. But our focus isn't on variability over time. At each time point, we were interested in how each indicator differs among the three groups. Therefore, two-way ANOVA was not used.

Results:

19. The results of PPT in figure 2 and figure 6 are the same. Please write all this results together and in 1 figure only.

20. Merge figure 3 and 7 and write their results in the same topic.

21. Merge figure 4 and 8 and write their results in the same topic.

22. Merge figure 5 and 9 and write their results in the same topic.

We changed the description in the methods section in accordance with your point, and we also changed the results in line with your point.

23. Line 399: “than the OA group” it is sed OA? Please correct

We corrected appropriately in line 331.

Discussion:

24. Line 418: “it suppressed”. It was alleviated, not supressed.

Sorry for the careless mistake. Correctly changed the sentence in line 351.

25. Lines 426-428: please reference the paragraph

Some of the sentences pointed out were inappropriate, we revised the sentences and referenced previous studies.

26. Line 434: Please clarify that PRIOR walking exercise did not alleviate, and highlight that you do not know if rats continue to exercise it could be different. This a limitation of the study.

As you point out, the lack of regular walking exercise after MIA injection is certainly one of limitation of our study. We added a reference to this point in line 369-372.

27. Why you did not verify anti-inflammatory markers through Western blotting? It would reinforces your PCR results. Even though you verified increased mRNA expression of anti-inflammatory markers, you do not know if it was sufficient to increase protein content. If possible, please do the Western blotting analysis from the same mRNA genes you verified.

Of course, it is best to analyze both mRNA and protein. However, the infrapatellar fat pad and synovium that can be harvested from the knee joints of a rat do not provide enough sample volume to perform both biochemical analysis and molecular biological analysis. Therefore, only mRNA was analyzed in this study.

28. Lines 473-475: please reference the paragraph

About lines 473-475 (line 394-396 in revised manuscript) reference, we added the reference No.24 and 25, and we corrected the uncertain statements.

29. Lines 486-487: “As mentioned above, the expression levels of IL-4 and IL-10 mRNA were increased by 6 weeks of the regular walking exercise”. It is repetitive, please delete. 

As you pointed out, we deleted the sentence.

30. Line 490: “the expression level of IL-1 mRNA”. Please correct to IL-1beta.

As you pointed out, we corrected the sentence in line 410.

31. Line 496: “was suppressed by the actions of IL-4 and IL-10”. You cannot infer this, once you did not verify protein content. In addition, you could only say it could be an explanation for these findings.

32. Lines 506-507: “Second, we were not able to examine the numbers of macrophages after the walking exercise”. Why you were not able?

As mentioned earlier, when infrapatellar fat pad and synovium are collected from a single rat, few synovium is left to perform immunohistochemical staining for macrophages. In this study, the number of rats used was minimized from the perspective of animal welfare, and therefore, we could not examine for macrophage alteration because we only used rats capable of analyzing IL -4 and IL -10 mRNA at the end of the 6-week exercise period.

33. Lines 517-518: “such as IL-6, tumor necrosis factor alpha (TNF-α), and nerve growth factor in the synovium”. Why you did not verified this genes and proteins? Please verify if it is possible. 

Again, the number of rats used in this study provides only enough infrapatellar fat pads and synovium for analysis of IL -4, IL -10, and IL -1 beta mRNAs. More rats need to be used to search for other cytokines. Therefore, it was considered as a limitation in this study.

34. Provide foundation sources

35. Provide authors contributions

It was not mentioned in the format of PLOS ONE, but we provided foundation sources and authors contribution in line with your point.

---

## [Decision Letter · Decision Letter 1]

29 Jun 2023

PONE-D-23-07563R1Regular walking exercise prior to knee osteoarthritis reduces joint pain in an animal modelPLOS ONE

Dear Dr. Sakamoto,

Thank you for submitting your manuscript to PLOS ONE. After careful consideration, we feel that it has merit but does not fully meet PLOS ONE’s publication criteria as it currently stands. Therefore, we invite you to submit a revised version of the manuscript that addresses the points raised during the review process.

Your manuscript, entitled "Regular walking exercise prior to knee osteoarthritis reduces joint pain in an animal model", has been reviewed. Your efforts to revise the manuscript are appreciated. However, the peer review process continues because Reviewer 2 has a few additional comments that the author should address. Please find the reviewer's commentary below.

We look forward to receiving your revised manuscript.

Kind regards,

Tanja Grubić Kezele, Ph.D., M.D.

Academic Editor

PLOS ONE

Journal Requirements:

Reviewers' comments:

Reviewer's Responses to Questions

**Comments to the Author**

1. If the authors have adequately addressed your comments raised in a previous round of review and you feel that this manuscript is now acceptable for publication, you may indicate that here to bypass the “Comments to the Author” section, enter your conflict of interest statement in the “Confidential to Editor” section, and submit your "Accept" recommendation.

Reviewer #2: All comments have been addressed

2. Is the manuscript technically sound, and do the data support the conclusions?

Reviewer #2: Yes

3. Has the statistical analysis been performed appropriately and rigorously? 

Reviewer #2: Yes

4. Have the authors made all data underlying the findings in their manuscript fully available?

Reviewer #2: Yes

5. Is the manuscript presented in an intelligible fashion and written in standard English?

Reviewer #2: Yes

6. Review Comments to the Author

Reviewer #2: Thank you for the correction and improvements in the manuscript. You did a very good job.

Walking exercise with treadmill:

In the next studies you could perform a incremental load test until exhaustion, it will give to you the max aerobic power based on the maximum velocity reached on the test. Than you can use a percentage from the maximum velocity and submit the rats to run in the treadmill based on individual test. You do not need more equipment, just a treadmill.

Results:

You did not mention on the Results the differences in the PPT between Sed + OA and Ex + OA, however it was one of the main conclusions of the experiment. Please make it clear in the results description.

Discussion:

The discussion is well written. However, it is too extensive. Please, be more direct to the point, shortening the discussion.

7. PLOS authors have the option to publish the peer review history of their article (what does this mean?). If published, this will include your full peer review and any attached files.

Reviewer #2: No

---

## [Author Response · Author response to Decision Letter 1]

21 Jul 2023

Tanja Grubić Kezele, Ph.D., M.D.

Academic Editor

We greatly appreciate you and reviewers’ specific comments and opportunity to resubmit with appropriate corrections. We revised our manuscript as reviewer2 pointed out. We have sorted out the descriptions of the considerations that were redundant. 

Each of your comments have been answered below.

Reviewer #2:

We greatly appreciate your comments that clarified the problems of the manuscript, and we are thankful for your thoughtfulness and kindness. We revised our second manuscript as you pointed out. Each of your comments have been answered below.

1. Walking exercise with treadmill: In the next studies you could perform a incremental load test until exhaustion, it will give to you the max aerobic power based on the maximum velocity reached on the test. Than you can use a percentage from the maximum velocity and submit the rats to run in the treadmill based on individual test. You do not need more equipment, just a treadmill.

Thank you for your advice. Based on your advice, we revised the manuscript on the study limitations.

2. Results:

You did not mention on the Results the differences in the PPT between Sed + OA and Ex + OA, however it was one of the main conclusions of the experiment. Please make it clear in the results description.

We apologize that we have not been able to make enough revisions to the results. We　have added the results the differences in the Sed + OA and Ex + OA groups in line with your points.

3. Discussion:

The discussion is well written. However, it is too extensive. Please, be more direct to the point, shortening the discussion.

Thank you for your advice. Based on your advice, we revised the manuscript on the discussion.

---

## [Editor Report · Decision Letter 2]

26 Jul 2023

Regular walking exercise prior to knee osteoarthritis reduces joint pain in an animal model

PONE-D-23-07563R2

Dear Dr. Sakamoto,

We’re pleased to inform you that your manuscript has been judged scientifically suitable for publication and will be formally accepted for publication once it meets all outstanding technical requirements.

Kind regards,

Tanja Grubić Kezele, Ph.D., M.D.

Academic Editor

PLOS ONE
---

## [Editor Report · Acceptance letter]

1 Aug 2023

PONE-D-23-07563R2 

Regular walking exercise prior to knee osteoarthritis reduces joint pain in an animal model 

Dear Dr. Sakamoto:

I'm pleased to inform you that your manuscript has been deemed suitable for publication in PLOS ONE. Congratulations! Your manuscript is now with our production department. 

Kind regards, 

on behalf of

Prof. dr. Tanja Grubić Kezele 

Academic Editor

PLOS ONE